# Bidirectional Relationship between Glycemic Control and COVID-19 and Perspectives of Islet Organoid Models of SARS-CoV-2 Infection

**DOI:** 10.3390/biomedicines11030856

**Published:** 2023-03-11

**Authors:** Tongran Zhang, Nannan Wang, Lingqiang Zhu, Lihua Chen, Huisheng Liu

**Affiliations:** 1Guangzhou Laboratory, Guangzhou 510006, China; zhangtongran@hust.edu.cn (T.Z.); d202280849@hust.edu.cn (N.W.); 2Department of Pathophysiology, School of Basic Medicine, Tongji Medical College, Huazhong University of Science and Technology, Wuhan 430074, China; zhulq@mail.hust.edu.cn; 3College of Life Science and Technology, Huazhong University of Science and Technology, Wuhan 430074, China; 4School of Biomedical Engineering, Guangzhou Medical University, Guangzhou 510180, China; 5School of Biomedical Sciences and Engineering, South China University of Technology, Guangzhou International Campus, Guangzhou 510006, China

**Keywords:** SARS-CoV-2, COVID-19, type 2 diabetes, hyperglycemia, glycemic control, islet organoids

## Abstract

Infection with severe acute respiratory syndrome coronavirus 2 (SARS-CoV-2) leads to morbidity and mortality, with several clinical manifestations, and has caused a widespread pandemic. It has been found that type 2 diabetes is a risk factor for severe coronavirus disease 2019 (COVID-19) illness. Moreover, accumulating evidence has shown that SARS-CoV-2 infection can increase the risk of hyperglycemia and diabetes, though the underlying mechanism remains unclear because of a lack of authentic disease models to recapitulate the abnormalities involved in the development, regeneration, and function of human pancreatic islets under SARS-CoV-2 infection. Stem-cell-derived islet organoids have been valued as a model to study islets’ development and function, and thus provide a promising model for unraveling the mechanisms underlying the onset of diabetes under SARS-CoV-2 infection. This review summarized the latest results from clinical and basic research on SARS-CoV-2-induced pancreatic islet damage and impaired glycemic control. Furthermore, we discuss the potential and perspectives of using human ES/iPS cell-derived islet organoids to unravel the bidirectional relationship between glycemic control and SARS-CoV-2 infection.

## 1. Introduction

The storm of coronavirus disease 2019 (COVID-19) has been sweeping the globe since the end of 2019. Infection with the new coronavirus, severe acute respiratory syndrome coronavirus 2 (SARS-CoV-2), leads to morbidity and mortality, with several clinical manifestations such as pulmonary failure, cardiovascular disorder, pancreatic and other organ dysfunctions, and has caused a widespread pandemic at an unanticipated level [1,2,3,4]. Severe outcomes can include respiratory failure and even death induced by viral pneumonia, although there is a large population that manifests with asymptomatic infections or mild upper respiratory tract disease. Infection with the virus occurs through several receptors, mainly the angiotensin-converting enzyme-II receptor (ACE2) [5,6,7], with the help of other cofactors such as TMPRSS2, NRP1 [8,9], transferrin receptor [10], and FURIN [5,11]. Following viral proliferation after infection, the immune responses are triggered to defend against the infection. Through multiple infection pathways by binding different receptors, the infection can cause damage to different organs, leading to various consequences. These include pneumonia and acute respiratory distress syndrome caused by lung damage, arrhythmia caused by cardiac damage, acute pancreatitis and diarrhea caused by gastrointestinal damage, hyperglycemia caused by pancreatic islet damage, and many long-term complications [12]. Although the majority of people with a normal immune system are able to defend against this viral infection, there are often complications accompanied by tissue damage. Notably, it has been shown that elderly people with chronic diseases have a higher risk of severe clinical symptoms and a higher mortality rate than younger people [4].

A bidirectional link between COVID-19 and diabetes mellitus has extensively been noted. On one hand, substantial clinical data have reported the increased severity and mortality in COVID-19 patients with new-onset diabetes since 2020 [13,14,15,16,17]. One-third of deaths that occurred in individuals with pre-existing type 1 diabetes (T1D) and type 2 diabetes (T2D) were caused by COVID-19 from 1 March to 11 May 2020 [18]. On the other hand, an increasing amount of evidence has shown that SARS-CoV-2 infection can induce the development of hyperglycemia and new-onset diabetes [19,20,21,22]. It is therefore of great importance to focus on the long-term influence of COVID-19 on glycemic control.

## 2. Method

The authors reviewed the state of knowledge of using islet organoid models to study the bidirectional relationship between SARS-CoV-2 infection and glycemic control. The keywords used for the search were “diabetes and SARS-CoV-2”, or “organoid models and SARS-CoV-2”, or “pancreatic islet and SARS-CoV-2”. More than 100 papers were identified using PubMed as primary source for the literature review.

## 3. Human Infection and Organ Injury

### 3.1. Virus Entry

In the early 21st century, a coronavirus named SARS-CoV caused a worldwide pandemic. As a beta-coronavirus, SARS-CoV-2 is highly similar to SARS-CoV. Both of them bind the ACE2 receptor and require other virus entry factors or proteases to facilitate the cell entry of these viruses [23,24]. Briefly, the S protein of SARS-CoV-2 is cleaved by proprotein convertases such as furin in the virus-producing cells to generate the S1 and S2 subunits. The S1 subunit then binds the receptor and the S2 subunit facilitates the attachment of the S protein to the virion membrane, thereby mediating membrane fusion. TMPRSS2 and cathepsin L are the two major proteases involved in the activation of the S protein and further enhance the attachment and endocytosis through which the viruses then invade the host cell [10].

### 3.2. Organ damage

The poor outcomes of COVID-19 are caused by not only damage to the lungs but also multi-organ damage, including to the lung, kidney, heart, liver, intestines, eyes, and skin [25]. Among these, the lungs have been thought to be the nidus and niche for viral proliferation. The spillage of the virus enters the bloodstream and then leads to the failure of other organs [25,26], although several studies have reported viral RNA reads in these non-lung tissues [27,28]. Early infection activates the immune system [29] and the acute phase always results in respiratory tract symptoms, fever, and other respiratory manifestations caused by uncontrolled inflammation, which is referred to as “cytokine storm syndrome” (SCC) [30,31]. To focus on this hyper-stimulated immune response, a large amount of cytokines and chemokines such as interferon γ [32], IL-1α, IL-1β, IL-2, IL-6, IL-10 [33], and ferritin [34] are involved in the resistance against the bacterial and viral infection, ultimately resulting in severe tissue damage and multi-organ failure, as well as predisposing the patient to complications of the disease [35]. COVID-19, at the level of pulmonary damage, can involve interstitial inflammation; diffuse alveolar damage (DAD), mostly bilateral DAD; and necrotizing bronchitis/bronchiolitis [36]. It has to be mentioned that COVID-19 patients can also suffer from acute fibrinous and organizing pneumonia, although the proportion is low [37]. Widespread manifestations have been shown to be a peculiarity of COVID-19, including those in gastrointestinal tissues [38]. Chai et al. demonstrated that not only the hepatocytes but also the intra-hepatic bile duct can express ACE2 receptors [39].

Moreover, the spread of SARS-CoV-2 from the site of the lung to the kidney through blood circulation can induce acute kidney injury because of the wide distribution of ACE2 receptors in the proximal tubules and podocytes [40]. In the case of cardiac disease in COVID-19, similarly to the aforementioned organs, the presence of ACE2 receptors on the cardiac myocytes facilitates infection with the virus [41]. Delorey et al. reported that both cell composition and gene programs have changed in the hearts of COVID-19 patients compared with those of healthy subjects. However, the extent of myocardial involvement might not be as severe as in other organs damaged by SARS-CoV-2 [28]. Nonetheless, there are recent clinical data that reveal the development of post-COVID-19 syndrome, the morbidity of which commonly includes anxiety, intermittent fever, headache, sleep disturbances, cognitive dysfunction, and other neurological illness [42,43].

### 3.3. SARS-CoV-2 Infection Leading to Islet Damage

The pancreas is composed of alpha, beta, delta, polypeptide (PP), mesenchymal, acinar, ductal, and endothelial cells, together with immune cells. Each of these has specific marker genes, for example, glucagon (GCG) for alpha cells, insulin (INS) for beta cells, PRSS1 for acinar cells, and somatostatin for delta cells, which play robust roles in the classification of pancreatic cells. ACE2 receptor and TMPRSS2 effector protease are the two acknowledged proteins participating in the entry of SARS-CoV-2 [32,38], and have been shown to be present in alpha, beta, delta, mesenchymal, acinar, and ductal cells [44,45,46]. Recent studies using pancreases from deceased patients and human pluripotent-stem-cell-derived islet organoids have both demonstrated that SARS-CoV-2 is able to infect both alpha and beta cells [47]. Müller et al. reported the existence of SARS-CoV-2 spike proteins in pancreatic alpha and beta cells from SARS-CoV-2-infected patients [48,49]. ACE2 was also detected in hPSC-derived beta cells and alpha that had been transplanted into SCID beige mice for 2 months [48]. Nonetheless, there are contradictions about the expression of ACE2 in pancreatic islets, especially beta cells. For example, ACE2 and TMPRSS2 were not enriched in single beta cells when human islet cells were analyzed [46], whereas a moderate signal of ACE2 was detected in endocrine cells, including beta cells [48]. A low expression level of N-protein in the beta cells from islets of the infected patients has been reported [48]. Thus, more research is needed to clarify the molecular mechanism underlying whether the SARS-CoV-2 infection of the islets would trigger diabetes [50].

The damage to beta cells caused by SARS-CoV-2 infection includes the depression of the beta cells’ insulin secretion and the induction of beta cell apoptosis, with a similar signaling pathway to T1D, as demonstrated from phosphoproteomic pathway analysis. Therefore, SARS-CoV-2 is thought to directly kill beta cells [10]. Single-cell RNA-seq further revealed that the expression of *INS* was downregulated in SARS-CoV-2-infected islets, while the expression of other marker genes such as *GCG*, *TM4SF4*, and *RFX6* in alpha cells, and *PRSS2* in acinar cells, were increased in infected islets [10]. As well as cell death, the de-differentiation or trans-differentiation of beta cells may be other mechanisms for the damage induced by SARS-CoV-2 infection; for example, the trans-differentiation of beta cells to alpha cells may elicit another cell fate induced by SARS-CoV-2 infection, resulting in abnormal glucose control [51]. As suggested by Shin, the impairment of the insulin/IGF signaling pathway may contribute to COVID-19 pathology through interferon regulatory factor 1 (IRF1) [52]. Although clinical evidence has shown that SARS-CoV-2 infection impairs pancreatic islet function, more research should be conducted to reveal the molecular and biochemical relationships between glucose metabolism and SARS-CoV-2 infection.

## 4. The Bidirectional Relationship between SARS-CoV-2 Infection and Glucose Metabolism

### 4.1. Diabetic Patients Are Prone to Infection by SARS-CoV-2 and Have Increased COVID-19 Severity

Increasing evidence has suggested that patients with pre-existing diabetes are prone to be infected by SARS-CoV-2, with a higher severity of COVID-19. For instance, Guan et al. reported that people with poor glycemic control caused by diabetes are more susceptible to SARS-CoV-2 infection, and that individuals with diabetes were shown to have a higher risk of severe COVID-19 [53]. To investigate the reasons why people with diabetes are more prone to infection, it is necessary to elucidate the relationship between glycol metabolism and the immune response. People with T2D and obesity are characterized by metabolic disorders combined with immune dysfunction, resulting in the accumulation and activation of immune cells located in different tissues, such as T cells, dendritic cells, macrophages, neutrophils, and B cells, which then enhance the production of chemokines and cytokines [54]. Other metabolic factors including lipokines, adipokines, and branched-chain amino acids are at an abnormal level in patients with T2D and obesity, which also contribute to inflammation [54,55]. The dysfunctional immune system is probably responsible for the higher risk of viral infection through the activation of the NLRP3 inflammasome [56].

A number of studies have reported that diabetes increases the risk of many infections. Interestingly, the more dysregulated the glycemic control in infected people, the higher the mortality and morbidity [57,58]. Since the outbreak of the pandemic, it has transpired that there is a higher risk of severe COVID-19 outcomes and higher rates of mortality in people with metabolic disorders, as suggested by case series from different areas [13,14,15,16,17]. A retrospective analysis from the USA has reported that 58% of the patients with COVID-19 who were admitted to ICUs had diabetes mellitus before being infected [59]. Case series in England also reported increased COVID-19 mortality in people admitted to the intensive care unit (ICU) or high dependency unit (HDU) who had T2D than in those without diabetes [60]. Alongside the severity and mortality, the risk of reinfections, vaccine breakthrough infections, and long COVID appear to be higher in patients with diabetes [55]. Lim proposed several mechanisms that link clinical severity and T2D, including inflammatory cytokine production, endothelial damage, insulin resistance mediated by the renin–angiotensin–aldosterone system, and so on, thus leading to series of complications such as thrombosis and other organ damage [61]. In addition, SARS-CoV-2 infection may result in the aggravation of metabolic diseases [16].

### 4.2. SARS-CoV-2 Infection Predisposes COVID-19 Patients to Hyperglycemia

Regardless of pre-existing diabetes before SARS-CoV-2 infection, it has been suggested that hyperglycemia is a characteristic clinical manifestation for people with severe COVID-19 [17,62,63]. An analysis of the Glytec database suggested a close association between hyperglycemia/hypoglycemia and poor outcomes of COVID-19 [64,65]. In the report by Bode et al., 257 of 1122 people with COVID-19 were diagnosed with poorly controlled hyperglycemia, with glucose readings higher than 10 mM. The risk of hyperglycemia and hypoglycemia, long COVID-19, and even death was shown to be higher in people with uncontrolled glucose levels [66]. Furthermore, the drug therapy for COVID-19 such as RNA polymerase inhibitors and chloroquine also appears to contribute to hyperglycemia [67,68]. New-onset diabetes has been reported in several studies, suggesting possible links between the onset of T1D, T2D, severe diabetic ketoacidosis, and SARS-CoV-2 infection [69,70]. More clinical evidence linking SARS-CoV-2 and glucose metabolism disorder is shown in Table 1, and the possible mechanism of this could be the aggravated inflammation leading to beta cell dysfunction, the impairment of the action of insulin, and a deterioration in insulin resistance [54]. Furthermore, the expression of ACE2 is downregulated after SARS-CoV-2 infection, and this plays a critical role in the renin–angiotensin system (RAS), thus leading to more severe symptoms [71,72,73]. An understanding of these mechanisms would help to investigate promising therapeutic approaches such as blocking the binding of ACE2 to the S protein of SARS-CoV-2.

## 5. Remaining Questions

Clinical data and basic research have established the bidirectional relationship between glycemic control and COVID-19. However, the details and the underlying mechanisms are largely unknown (Table 2). ACE2 is expressed in the pancreas, with evidence showing that both the endocrine and exocrine function of the pancreas can be affected by SARS-CoV-2 infection [83], but its distribution and abundance in endocrine and non-endocrine cells is yet to be clarified. To address this question, multiple pancreatic cell types including delta cells, PP cells, acinar cells, and ductal cells should be generated from hPSCs, followed by cell sorting strategies to further investigate the distribution and abundance of ACE2 in each cell type. The dynamics of ACE2 during pancreas development are still unknown. To date, there is poor knowledge about the expression of ACE2 during the development of the human pancreas. Current islet organoid differentiation methods could mimic the natural development of the human pancreas and thus allow us to monitor the expression of ACE2 across the development of the pancreas. Addressing this question would help us understand the impact of SARS-CoV-2 infection on the development of the pancreas. Another remaining question is how the SARS-CoV-2 infection influences the pancreatic endocrine cells’ function and plasticity, and what the underlying mechanisms are. Clinical evidence has suggested that beta cell de-differentiation and trans-differentiation occur upon infection with SARS-CoV-2, but the details and mechanisms are not clear. An investigation into the molecular mechanisms of pancreatic dysfunction is in progress, for example, the activation of the Na+/H+ exchanger in the pancreas, leading to cell damage [84,85]. Moreover, inflammatory cytokines and molecular mimicry may also be correlated with pancreatic cell damage [85,86]. Clinical data have shown that SARS-CoV-2 infection can result in hyperglycemia and/or diabetes (Table 1). It is not clear whether the infection resulted in T1D or T2D, and, if so, whether the effect is reversible or not. Although a connection between T1D and viral infections has been reported [87], more cohort studies and experimental research based on a disease model are expected to solve this question. It remains to be discovered whether the severity of impaired glycemic control correlates with the virulence of SARS-CoV-2 variants. There has been poor evidence showing this correlation, and thus islet organoids could be an unlimited source for investigating this question. Monitoring the function of organoids and their glucose sensitivity as affected by different SARS-CoV-2 variants may be a practical solution. Moreover, we need to clarify multi-organ interactions in the context of SARS-CoV-2 infection and whether islet damage would also affect other organs. Furthermore, it would be valuable to investigate whether beta cells are infected by SARS-CoV-2 simultaneously with the lungs, or whether they are affected following damage and inflammation to other tissues. Figure 1 presents the applications of multi-organoid systems for addressing this question. Recent progress was made by Tao et al. in the successful establishment of a human liver–islet system [88], and further attempts in this field will help to address these unsolved questions.

## 6. Strengths and Challenges of Using Islet Organoid Models

As mentioned above, it has been reported in numerous studies that diabetes is a high-risk factor for SARS-CoV-2 infection (Table 1). Severe symptoms, increased comorbidities, and higher mortality in people with COVID-19 were shown to be associated with pre-existing T1D and T2D [77]. SARS-CoV-2 infection, on the other hand, can induce hyperglycemia, although the mechanism is still unclear [66].

In order to investigate the impacts of SARS-CoV-2 infection on the pancreas, human pluripotent stem cell (hPSC)-derived islet organoids have been used as a valuable model. Organoids are typically 3D tissue models derived from hPSCs or expanded islets extracted from human samples. These are self-arranged and characterized with a measure of complexity and organ-like cellularity as well as functions to a certain extent [89,90]. The advantages of organoids include their tissue-like structures and the involvement of cell–cell interactions; hence, they are thought to have authenticity for modeling viral infections and accelerating the discovery of antiviral therapeutics. The islet organoids derived from stem cells in vitro have been instrumental sources for understanding the pathogenesis of diabetes and pancreatic dysfunction [91,92]. By using this model, researchers have found the implications of transcription factor GATA6 for diabetes [93]. Moreover, studies using this model reported the acquirement of mature function in iPSC-derived beta cells in vivo and validated the capacity of therapies for diabetes through the transplantation of islet organoids under the kidney capsule [92] or in macro-device implants [94]. Consistent with human primary beta and alpha cells, hPSC-derived beta and alpha cells also express the SARS-CoV-2 spike protein receptor ACE2 and permit viral invasion and robust replication, suggesting the potentiality of islet organoids to represent viral infections [95]. As a proof-of-concept experiment, islet organoids and several other organoids were used to model SARS-CoV-2 infection in various human organs [90,95]. Firstly, the islet organoid model of SARS-CoV-2 infection showed increased insulin resistance and chemokine expression, and decreased endocrine functional pathways caused by increased apoptosis, resulting in insulin deficiency and even the onset of diabetes in some COVID-19 patients [95]. Moreover, by using human liver ductal organoids as models, Zhao et al. recapitulated the infection, transmission, and evolution of SARS-CoV-2 [96]. In addition, drugs, such as mycophenolic acid, can be identified by inhibiting the entry of SARS-CoV-2 in lung organoids derived from pluripotent stem cells as models [97]. Moreover, Lai et al. modeled SARS-CoV-2 infection with hiPSC-derived retinal organoids and found the expression of ACE2 and TMPRSS2 [9]. Moreover, a recent study modeled SARS-CoV-2 infections in hiPSC-derived kidney organoids and reported the direct kidney cell infection and subsequent kidney injury induced by SARS-CoV-2 infection [98]. To promote a further understanding of SARS-CoV-2 infection and the development of novel therapeutics using organoids as models, omics/proteomics analyses could first be performed to investigate the effects of viral infection on the expression profile. Western blot, immunocytochemistry, and other molecular techniques could help to identify the distribution of the entry factors of the virus. Glycemic control and islet cell damage in SARS-CoV-2-infected islet organoids could then be assessed both in vitro and in vivo, which would help us to further understand the mechanism of infection and define a road map for new therapeutics (Figure 1). The intravital imaging and microscopic techniques of islet organoids also allow the exploration of dynamic changes after transplantation into mouse models [99].

Nevertheless, our understanding of the relationship between glycemic control and COVID-19 is still elusive, partly because of the limitations of the current islet organoid models. There are still several obstacles to establishing an authentic islet organoid model that could best mimic the primary human islets. First, the human pancreas contains multiple cell types, while the current islet organoids mainly contain alpha and beta cells, and the ratio and distribution and cell–cell interactions are random. Second, because of the incomplete developmental systems, the cells of organoids are immature and may be characterized by different cell properties. Third, the lack of blood vessels as well as sympathetic nerves could also lead to disordered cell arrangements and intra-islet signal transduction. In addition, high heterogeneity is another limitation in the field of stem cell regeneration, which has been reduced by the construction of micro-well platforms and bio-printing, as well as by robust differentiation [100]. Furthermore, relatively low levels of oxygen and nutrients, and physiological and physical support lead to cell death, necrosis, and apoptosis. Lastly, the use of islet organoid models is also restricted by the lack of an immune system. Moreover, the expression of ACE2 is relatively low in endocrine compartments compared with exocrine compartments. Evidence has shown that the pancreatic ducts and microvasculature are more likely to be targeted in SARS-CoV-2 infection [5,46]. Therefore, whether islet organoids are an ideal SARS-CoV-2 infection model remains to be observed and determined.

Although there are plenty of advantages of using islet organoids, the existing differentiation methods are not advanced enough to induce the full range of cell types. Incomplete cell types, unstable induced differentiation, organoids that are not fully functional, erroneous cell proportions, and the lack of blood vessels, an immune system, and sympathetic nerves are regarded as the challenges, which are worth overcoming so that hESC-derived organoids could function better when modeling SARS-CoV-2 infection and COVID-19-related dysfunction. For instance, as a safe and convenient system, organ-on-a-chip technology can identify the cross-talk between different SARS-CoV-2-infected organs and elucidate the mechanisms underlying the elevated mortality in diabetic COVID-19 patients. More detailed improvements will be discussed in the next section.

## 7. Conclusions and Prospects

Since the outbreak of COVID-19, the application of organoids for modeling of the disease has rapidly progressed. Nevertheless, islet organoid models should be further optimized by additional explorations to better unravel the relationships between glycemic control and infectious diseases [89].

The prospective solutions for the improvement of islet models are recapitulated in Figure 1. Firstly, differentiation methods should be improved for the directed generation of other pancreatic cell types, including both endocrine and exocrine cells. FGF family member 7 (FGF7) not only enhances the expression of PDX1, thereby promoting the differentiation of beta cells [101], but may also regulate the differentiation of pancreatic exocrine cells [102,103]. As well as FGF7, other FGFs may have a different potential to induce other pancreatic cell types, so they need to be surveyed. Moreover, hepatocyte nuclear factor 6 (Hnf6) promotes the differentiation of duct cells and also regulates acinar cell development [104,105,106]. Furthermore, histone deacetylase (DHAC) inhibitors also play essential roles in amplifying the endocrine progenitors of NGN3+ and ductal differentiation, but abolish the differentiation of acinar cells [107,108]. In addition, cell purification can be used to reconstruct islet organoids with the affiliated vessels and nerves. CD49a and DPP4 have been identified as surface markers for beta cells and alpha cells, respectively [103,109,110]. Hence, they can be used to efficiently purify beta cells and alpha cells after the elimination of irrelevant cells through a magnetic sorting strategy [103,109]. In this regard, one task would be to identify the cell surface biomarkers for other pancreatic cell types. The purified cell types can be used to clarify the presence of ACE2 and other related factors in the pancreas. They can also be further used to study the transcriptomics and proteomics related to viral entry and infection. Analyses of the transcriptome profile, multi-omics, and organoids with genome editing can further reveal the viral pathogenesis involving alterations in a series of molecular and cellular events, for example the up- or downregulation mechanism of ACE2 after infection, and the investigation of other virus entry factors [73].

Moreover, animal models should be established in order to study the long-term effects of infection and/or impacts on diabetic subjects. For example, hACE2 transgenic mice [111,112] can be used to induce a T1D model with streptozotocin (STZ) [113] or a T2D model with a high-fat diet (HFD) [114]. After hACE2 mice that have been implanted with islet organoids are infected with SARS-CoV-2, the physiological and pathological changes in the pancreas can be observed to investigate the damage and dysfunction of pancreatic cells caused by viral infection. The transcriptomic profile and omics would help in analyzing the modification of genetic transcription after infection, which would lead to a further understanding of the mechanism of how the infection causes impaired glycemic control. Moreover, strengthening the techniques of multi-organ chips, with the layout/interaction of multiple organs such as islets, lungs, intestines, livers, and immune cells, will more closely resemble human physiology and accelerate the resolution of unsolved problems [88,115]. For example, Tao et al. successfully developed a novel microfluidic multi-organoid system replicating the human liver–islet axis in both healthy and diseased states [88]. This technology would allow the remaining questions such as multi-organ interactions after SARS-CoV-2 infection and future drug discovery to be resolved.

Islet organoids enable us to investigate the mechanisms of the association between diabetes and the poor outcomes of COVID-19. Considered as an unlimited platform, islet organoids are expected to have high potential for the study of viral infections, COVID-19 disease therapy, and therapy for other diseases such as diabetes. We believe that the use of islet organoids would be a breakthrough in the study of SARS-CoV-2 infection and COVID-19 treatments.

## Figures and Tables

**Figure 1 biomedicines-11-00856-f001:**
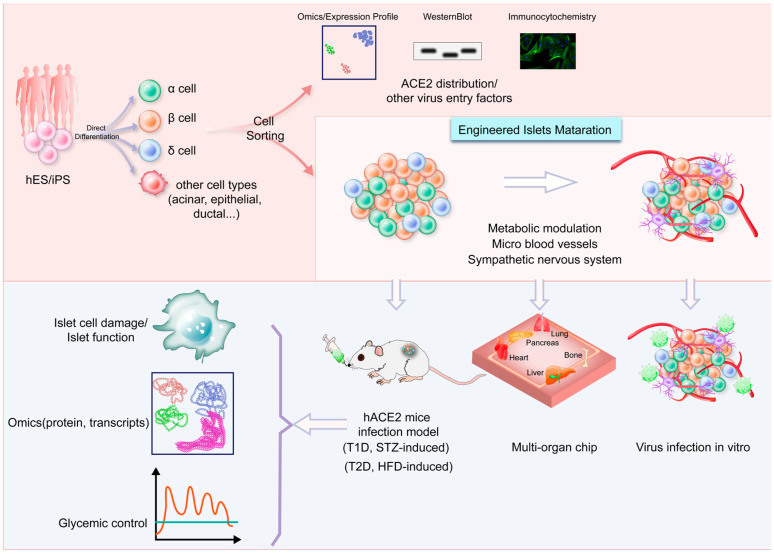
Islet organoids for unraveling the relationship between glycemic control and SARS-CoV-2 infection. The generation and purification of pancreatic cell types would help clarify the distribution and abundance of ACE2 and other related factors in the pancreas. They would also allow the engineering of human islet organoids that are physiologically similar to primary human islets. Mature human islet organoids can be used for studies of viral infections and multi-organ chip assessments. The identification of islet dysfunction and glycemic control in SARS-CoV-2-infected normal islet organoids in healthy or diabetic mouse models would help unravel the bidirectional relationship between glycemic control and SARS-CoV-2-infection.

**Table 1 biomedicines-11-00856-t001:** Clinical studies showing the bidirectional relationship between glycemic control and SARS-CoV-2 infection.

	Region	Source of Data	Main Findings	Reference
STUDY 1	England	13,809 patients admitted for COVID-19 to the HDU/5447 admitted to the ICU Mean age: 70/58	34.7% mortality with T2D; 25.5% mortality without T2D	[60]
STUDY 2	England	23,698 COVID-19-related deaths (with and without diabetes)	31.4% mortality with T2D; 1.5% mortality with T1D	[18]
STUDY 3	Korea	5307 people with COVID-19	Increased severity and higher mortality in 14.5% individuals with T2D	[74]
STUDY 4	USA	395 patients with COVID-19 Age: 18–35	3.8% mortality without comorbidity; 13.6% mortality with diabetes (deceased); 18.5% mortality with diabetes (diagnosed)	[75]
STUDY 5	Mexico	757,210 patients with COVID-19	Patients with diabetes had a 49% risk of death higher than those without diabetes; Diagnosis of T2D as COVID-19 outcome in both young and old	[76]
STUDY 6	USA	1544 patients with COVID-19	Hyperglycemia and hypoglycemia both contribute to poor outcomes of COVID-19	[64]
STUDY 7	China	7337 with COVID-19 Ages: 18–75	13% of patients with T2D; Death rate is 1.49-fold higher in the T2D cohort	[77]
STUDY 8	China	1099 patients 639 male/460 female Mean age: 47	Individuals with diabetes are more susceptible to SARS-CoV-2 and more easily develop a severe course of COVID-19	[53]
STUDY 9	England	Population-based cohort study	COVID-19-related mortality increases in people with a higher glycosylated hemoglobin level	[78]
STUDY 10	France	2,608 patients with COVID-19 Age: 56.0 (±16.4)	Patients with T1D (age > 65–75) had higher rates of COVID-19-related mortality	[79]
STUDY 11	China	92 patients with COVID-19 without metabolic-related diseases	New-onset insulin resistance, hyperglycemia, and decreased HDL-C in these patients	[80]
STUDY 12	Italy	551 patients with COVID-19 344 male/207 female Age: 61 ± 0.7	46% overt hyperglycemia; 12% new-onset diabetes; glycemic abnormalities last for 2 months after recovery	[81]
STUDY 13	UK	30 children with new-onset T1D Age: 23 months–16 years	The number of children with new-onset T1D increased since the COVID-19 pandemic. Some of these patients had been infected/exposed to SARS-CoV-2	[20]
STUDY 14	France	1 woman with COVID-19 and a history of gastric bypass Age 29	The COVID-19 patient was diagnosed with new-onset diabetes after 1.5 months	[82]
STUDY 15	Italy	413 patients with COVID-19	21 of 413 (5.1%) had new-onset diabetes; Patients with new-onset diabetes reported higher severity and mortality than those with pre-existing diabetes	[63]

**Table 2 biomedicines-11-00856-t002:** Remaining questions and proposed solutions (* indicates progresses that have been made by researchers).

	Remaining Questions	Proposed Solutions/Progress *
1	The distribution and abundance of ACE2 in endocrine and non-endocrine cells	Generating missing pancreatic cell types, biochemical analyses
2	The dynamics of ACE2 during the development of the pancreas	Biochemical analyses
3	The mechanisms of how SARS-CoV-2 infection influences the pancreatic endocrine cells’ function	Transplantation of islet organoids Activation of the Na+/H+ exchanger * Inflammatory cytokines *
4	Whether infection results in T1D or T2D	More cohort studies and experimental research based on disease models
5	The correlation between impaired glycemic control and the virulence of SARS-CoV-2 variants	Infection model of islet organoids
6	Multi-organ interactions after infection?	Multi-organoid systems, co-culture systems

## Data Availability

All data for the review are available within the article.

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
