# Peer review of "Bidirectional Relationship between Glycemic Control and COVID-19 and Perspectives of Islet Organoid Models of SARS-CoV-2 Infection"

_biomedicines, 2023, doi:10.3390/biomedicines11030856_

Round 1

Reviewer 1 Report

General comments:

The authors review the literature of the impact of COVID-19 on biological, pathological, and clinical parameters of diabetes. This is timely and important as the world treat the long-term complications of COVID-19, including diabetes and progressing diabetes in already diabetic patients.

The biological, clinical and pathological impact are well-described.  suggest expanding on the topic of using organoids to study diabetic disease in general prior to introducing its use in modeling COVID-19 induced diabetes.

Major changes:

In paragraph 6. Line 210, please expand on the use of organoids to study diabetes in general https://pubmed.ncbi.nlm.nih.gov/36714557/, https://pubmed.ncbi.nlm.nih.gov/36550929/, https://pubmed.ncbi.nlm.nih.gov/36060810/, and many more examples. Then expand on using these organoids for SARS-COVID19.

I suggest describing the type of analyses to be performed on the organoids in order to study and develop novel therapeutics.

I suggest adding a specific paragraph expanding on the limitations and how to overcome these limitations in using organoids to model islet COVID-19 dysfunciton.

Minor changes:

Page 6, line 217 – the definition of Organoids as deriving exclusively from hPSCs can be disputed by some researchers. Some organoids are derived by extracting islets from pancreatic samples and expanding them in Matrigel.

Reviewer 2 Report

The relationship between glycaemic control and SARS-Cov2 infection is reviewed both from the point of view of diabetes being a predisposing factor for severe infection and from that of potential damage to glucose homeostatic mechanisms resulting from infection.  This is a very interesting and topical area and, given the appreciation of diabetes and its links to other disorders, such as Alzheimer’s, will generate a large readership.

There are a number of areas of the review which require further attention from the authors. 

1) The piece seems to have been given the wrong title.  Organoids have a prominent position in the title but the review was about two thirds of the way through before there was any mention of pancreatic organoids at all.  Given the weight authors have given to describing the relationships between glycaemic control and COVID-19, I would suggest an amendment of the title.

2) Section 4: “Remaining questions”

This section is out of place in a scientific review.  It is not enough to list potential questions.  The purpose of a review is to inform the reader of the state of knowledge of a particular topic.  This section should be re-written to outline how the questions raised are being addressed by researchers and what progress has been made.

3) Section 5: “Strengths and Challenges…”

Paragraph 2 of this section ought to be much expanded.  The reader should be informed of what previous organoid modelling of SARS-CoV-2 infection has told us.  It is not enough to list the attempts that have been made, studies should also be evaluated.

4) Table 1 is useful and informative.  It does require some re-phrasing, however. I’m assuming that “deaths” referred to for study 1 and study2 relate to the mortality rates of SARS-CoV-2 patients with and without diabetes.  That is not what has been stated in the table and it should be clarified.

5) The English language needs extensive editing, preferably by a native speaker.  Some sections are quite difficult to follow and the meaning and impact are lost due to poor English.  There is some confusion over tense (including in the very first line of the Abstract) and inappropriate and incorrect usage of idioms.

Reviewer 3 Report

The reviewer would like to raise the following points for the authors to consider:

 Abstract:

·         Line15: Remove the abbreviation “T2D”. It is better to insert it in line 50 (introduction).  

 Introduction

·         Line 42: The authors should briefly explain the consequences of organ damage.

·         Line 46: Insert the reference.

·         Line 50: Enter the abbreviation “T1D and T2D”.

 Before starting with the paragraph “Human infection and organs injury”, the authors must insert a section with “Methods.” In particular, they must be answered to these simple questions:

·         What kind of revision did they want to write?

·         What keywords did they use for the search?

·         What National Library they consulted for the review of literature?  (PubMed, Scopus, Academia....).

 You can take inspiration from the examples below.

doi: 10.1016/j.ijid.2021.11.009

doi: 10.3390/ijerph16204031

 Human infection and organs injury

Organ damage

·         Line 87 to 88: Remove the sentence “According to ……. disease”. It does not add value to what has been said previously.

·         Line 94: Indicate Delorey et al. as previously done for Chai et al.

 Bidirectional relationship between SARS-CoV-2 infection and glucose metabolism

Diabetic patients are prone to be infected by SARS-CoV-2 and have increased severity of COVID-19

·         Line 141: Authors should better introduce the paragraph.

·         Line 162: Define the abbreviation “ICU” - “HDU”

 Remaining questions

·         Line 195 to 208: Authors might include a table with questions. It might be easier and more attractive to read.

 Strengths and challenges of using islet organoid models.

·         Authors should provide more details regarding the strengths and challenges of using organoid models. This should be the heart of the review. Argue better how much is present in the literature.

 Conclusion

·         Remove the bulleted list and try to make a more homogeneous speech. The authors could divide the paragraph into two parts:  

o   Future prospectives

o   Conclusion

Round 2

Reviewer 2 Report

This is an interetsing and topical study.  The paper now reads well and I have been impressed by the revisions made by the authors.

Reviewer 3 Report

The authors have satisfied all the objections. Now the work is ready to be published